# Dynamics of Spatiotemporal Variation of Groundwater Arsenic Due to Salt-Leaching Irrigation and Saline-Alkali Land

Shuhui Yin [1,2,†], Yuan Tian [1,2,†], Linsheng Yang [1,2] , Qiqian Wen [1,2] and Binggan Wei [1,2,*]

1   Key Laboratory of Land Surface Pattern and Simulation, Institute of Geographic Sciences and Natural Resources Research, Chinese Academy of Sciences, Beijing 100101, China
2   College of Resources and Environment, University of Chinese Academy of Sciences, Beijing 100049, China
*   Correspondence: weibg@igsnrr.ac.cn; Tel.: +86-(0)-10-64856504
†   These authors contributed equally to this work.

**Abstract:** Determining the link between the evolution of salt-leaching irrigation, saline-alkali land, and groundwater arsenic (As) is essential to prevent groundwater arsenic pollution and implement appropriate soil salinization control projects. The objectives of our study were to explore the spatiotemporal correlation of saline-alkali land and salt-leaching irrigation with groundwater As in the Hetao Plain. Therefore, groundwater As concentrations during Period I (2006–2010) and Period II (2016–2020) were collected by historical data and chemical measurements. Salt-leaching irrigation area and saline-alkali land area in Period I and Period II were extracted through remote sensing data. With the increase of the salt-leaching irrigation area level (SLIAL) and saline-alkali land area level (SALAL), the variation trend in groundwater As concentration slightly fluctuated, with an increase in the SLIAL at the low SALAL, which may be because short-term flooding may not considerably enhance As mobilization. Lower groundwater As concentrations appeared in regions with higher SLIAL and lower SALAL. A larger saline-alkali land area (higher SALAL) increased the groundwater As concentration. The path analysis model confirmed that salt-leaching irrigation may increase groundwater salinity to affect groundwater As levels and to decrease the saline-alkali land area. From Periods I to II, the difference in path analysis results may imply that the decrease in the saline-alkali land area may have influenced As mobilization due to competitive adsorption caused by the increase in total dissolved solids (TDS) in groundwater. Our results provide new insights for the impacts of saline-alkali land and salt-leaching irrigation both on groundwater As concentration and the geochemical processes of As enrichment in arid and semi-arid areas with more serious salinization.

**Keywords:** saline-alkali land; salt-leaching irrigation; groundwater; arsenic; spatiotemporal variation

## 1. Introduction

Soil salinization has become an urgent global environmental and socioeconomic issue, especially in arid and semi-arid areas [1]. Poor drainage may generate soil salinity problems because capillary action causes the rise of the water table to accumulate salts in the soil near the root zone of the plants in the agricultural irrigation system [2]. The Hetao Plain is a representative salinization-irrigated district that is affected by serious soil secondary salinization caused by inappropriate irrigation, poor drainage, and high groundwater levels [3–5]. Land use patterns, the texture, the structure, and the capacity of the soil can affect groundwater quality [6,7]. Urbanization affects the hardness of groundwater, where soft groundwater is characterized by agricultural areas and hard groundwater is characterized by a mineralized subsoil in the state of Morelos, Mexico [6]. Aquifers were susceptible to nitrate contamination in the area localized to most agricultural activity in Flamouria Basin in Northern Greece [7]. Saline-alkali land has markedly higher concentrations of total dissolved solids (TDS), $Na^+$, and $HCO_3^-$ in groundwater than other land types in Shuangliao City, Northeast China [8]. In autumn, intensive

flooding irrigation is used to leach salt from the soil to maintain the soil salinity within a reasonable range in the Hetao Plain [3,5]. Intense short-term flood salt-leaching irrigation in autumn typically results in significant groundwater recharge, raising the water table [9]. Soil salts can be leached into groundwater or transferred to neighboring areas without irrigation due to significant lateral groundwater gradients [3]. Soil salt-leaching into groundwater led to an increase in the groundwater TDS content and enhanced groundwater salinization [5,10,11]. Moreover, salt-leaching irrigation in saline-alkali land elevates the concentrations of $HCO_3^-$, $SO_4^{2-}$, $Ca^{2+}$, $Na^+$, and $Cl^-$ in the groundwater [12]. In addition, the rise in the water table and TDS concentration in the groundwater also affects the distribution and evolution of saline-alkali land [3,5,13,14]. Thus, the salinization of groundwater and soil is interrelated. The distribution of saline-alkali land and salt-leaching irrigation may change the hydrogeochemical properties of groundwater. It is important that the relationship between irrigated soil quality and trace elements' concentration in the groundwater should be investigated in the agricultural irrigation area [15].

Chronic arsenic (As) exposure induces several types of cancer (e.g., lung and bladder cancer) [16], endothelial dysfunction [17], impairment of intestinal stromal cells [18], chronic kidney disease [19], hypertension, and skin lesions [20]. The World Health Organization (WHO) recommends 10 μg/L as the maximum permissible concentration of As in drinking water [21]. The Hetao Basin is a typical region with geologically high As ( >10 μg/L) in groundwater, which has affected the public health of 300,000 people for decades [22–24]. In the Hetao Basin, the release of As into groundwater is usually influenced by the reductive dissolution of Fe(III) oxides through the degradation of organic matter and the oxidation of sulfide ($H_2S$), as well as As desorption caused by an increase in pH and the coexistence of anions, such as $HCO_3^-$ and $PO_4^{3-}$ [25]. Groundwater with high As concentrations is accompanied by low concentrations of $NO_3^-$ and $SO_4^{2-}$ and high concentrations of dissolved Fe, Mn, $HCO_3^-$, P, and $S^{2-}$ in the shallow alluvial-lacustrine aquifers in the Hetao Basin [22]. The coexistence of high As and saline groundwater was observed in the Hetao Basin [24,26]. Salinization processes may influence the mobilization of As in groundwater [10,11,24]. This can be attributed to the simultaneous As enrichment and $SO_4^{2-}$ reduction in the reductive aquifer when the TDS concentration is below 3000 mg/L [24]. The increase in salinity may promote flocculation of iron oxides, which can limit As enrichment in groundwater [10,27]. However, increasing the salinity and bicarbonate in groundwater can promote the release of As via competing adsorption in aquifers [28,29]. In general, salt-leaching irrigation can transfer soil salt into groundwater, which affects the distribution of saline-alkali land, the water table, the salinity, and related salt ion concentrations of groundwater [3,5,8]. Subsequently, the release of As into groundwater may be limited or promoted owing to the changes in the water table, the salinity, and related salt ion concentrations of groundwater [10,11,24]. Therefore, we assume that spatial and temporal distribution of As in groundwater may be associated with the distribution of saline-alkali land and salt-leaching irrigation. Exploring the effects of saline-alkali land and salt-leaching irrigation on groundwater As may be helpful to prevent groundwater arsenic pollution and to better implement appropriate soil salinization control projects.

Remote sensing satellite image data can easily characterize the macroscopic changes of the surface environment in a large geographical area, including land use (also includes evolution of wetlands and saline-alkali land), active faults, rivers, and vegetation types [8,30–34]. Correlatively, the impacts of wetland evolution, active faults, and the distance from the river on groundwater As concentration have been clearly explained [30–32]. Studies on soil salinization, salt-leaching irrigation, and groundwater chemistry have mostly focused on the microscopic scale, including the transport of soil salt ions between soil and groundwater due to salt-leaching irrigation and the distribution relationships of groundwater depth, salinity, and soil salt content [3,12,14,35]. However, these studies were mainly based on chemical experiments, and the effects of soil salinization and salt-leaching irrigation on groundwater chemistry at a larger scale cannot be economically and rapidly characterized. Our study explores fresh insight into the quantitative relationship between

the evolution of salt-leaching irrigation and saline-alkali land at the macroscopic scale based on remote sensing data with spatiotemporal variation of groundwater As in the Hetao Plain, which can be used at a larger geographical scale and even a global scale.

The objectives of this study were to: (1) characterize the spatiotemporal variation of groundwater As in shallow aquifers, (2) determine the spatiotemporal distribution of salt-leaching irrigation areas and saline-alkali land using remote sensing imagery, and (3) assess the quantitative relationship between salt-leaching irrigation, saline-alkali land, and groundwater As.

## 2. Materials and Methods

### 2.1. Geological and Hydrological Setting

The study area is the flat plain of the Hetao Basin, the Hetao Plain, which is located in the arid and semi-arid areas from the south of the Langshan Mountains, adjacent to the north of the Yellow River. The study area has a total area of 8949 km$^2$ from the east of the Ulanbuh Desert to the west of Wuliangsuhai Lake (excluding Ulanbuh Desert) (Figure 1). The average annual precipitation ranges from 130 to 220 mm, while the annual evaporation ranges from 2000 to 2500 mm [24]. The elevation is in the range of 1021–1100 m.

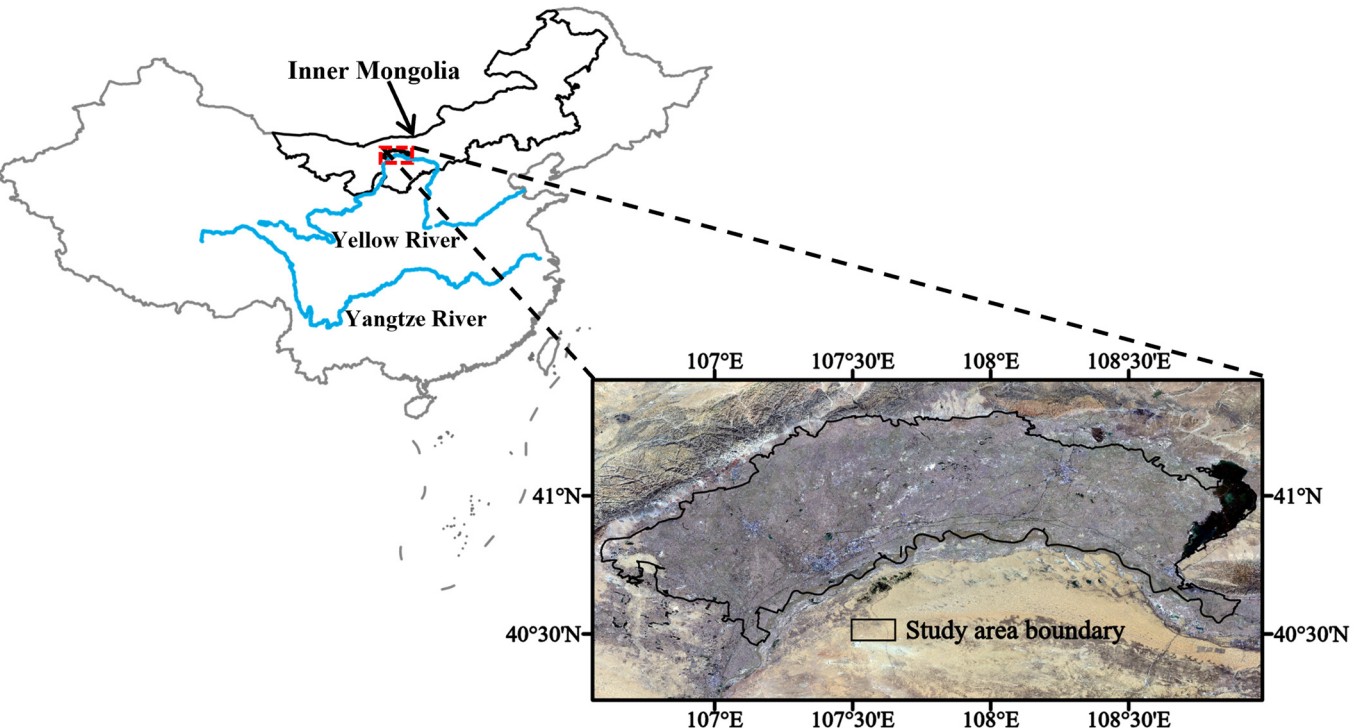

**Figure 1.** Geographical site and Landsat true-color composite images of Hetao Plain.

Hetao Basin, located in the northern margin of the Ordos Syncline of the North China Platform, is regarded as one of the Cenozoic rift basins. Tectonic movement and the paleo-climate affect the sedimentary environment, paleo-geography, and lithology. Great amounts of salinity (such as calcite and gypsum) were accumulated due to the occurrence of red or deep-brown sediments in an oxidative environment during the Tertiary period. Inland lacustrine sediments had locally been deposited and thick Mid-Cenozoic sedimentary formation had developed in relation to the closed geologic structure and continuous tectonic subsidence during the Quaternary period [22,36]. Fluvial and lacustrine sediments with silt and fine sand were deposited during the Quaternary period in the Hetao Plain [37].

The aquifers overlying clay layers at a depth of 40 m (<40 m BLS) are regarded as shallow groundwater [25]. The shallow aquifers are generally multi-layered, varying from unconfined to leaky-confined in the shallow deposits, which is characterized as an

interbedding of silt and fine sands rich in organic matter with a relatively low hydraulic gradient and groundwater flow rates [22,36,37]. The north area is the weak runoff area with lower permeability due to higher clay particle content, and the south area near the Yellow River is the strong runoff area with higher permeability [38]. The depth of the water table generally changes from 2 to 20 m multi-yearly, affected by Yellow River irrigation in Hetao Basin, and the depth of the water table is approximately 1 m BLS in the flat plain [35,36]. The recharge to shallow groundwater includes irrigation channels, rainfall infiltration, and surface water, while the discharge paths include evapotranspiration, drainage, and pumping in the study area [29,37]. The flow direction of shallow groundwater is generally from northwest to southeast and north, and from southwest to north and south [29,30,38].

### 2.2. Groundwater as Dataset

Shallow groundwater is more vulnerable to saline soil leaching contamination [10]. Aquifers above the clay layer host shallow groundwater (<40 m BLS), including alluvial-lacustrine aquifers and alluvial-pluvial aquifers [23,25,37]. Therefore, we mainly selected groundwater samples below 40 m for analysis. Groundwater data, including As concentration, coordinates, well depths, and sampling times for each sample, were collected from historical documents [22,23,30,35,37,38]. These shallow groundwater samples were collected in the Holocene and late Pleistocene aquifers. The lithology of the aquifers was mainly composed of silt and fine sand and the aquifers were above the clay layer. Therefore, all sampled sites can be affected by the surface recharge. Samples were collected between July and September in 2006, 2007, 2008, 2009, 2010, 2016, and 2017. Moreover, the contents of trivalent As as (As (III)), total dissolved solids (TDS), pH, $Na^+$, $Cl^-$, $HCO_3^-$, and $SO_4^{2-}$ in shallow groundwater samples were collected. In addition, 73 shallow groundwater samples (<40 m BLS) collected in August 2020 were analyzed. TDS content and pH were measured with a portable water quality instrument. Arsenic concentration was tested by inductively coupled plasma mass optical emission spectroscopy (ICP–OES). The $SO_4^{2-}$ concentration was determined by ion chromatograph. Interannual variations in groundwater As from 2006 to 2010 were not significant [23]. From 2011 to 2016, water-saving reconstruction was widely applied in the irrigation area of the Hetao Plain. This water-saving reconstruction markedly decreased the groundwater table [39], possibly impacting the distribution of saline-alkali land and the concentration of As in groundwater. Therefore, the groundwater As data were divided into two periods for analysis: 2006–2010 (Period I) and 2016–2020 (Period II) (Figure 2).

### 2.3. Remote Sensing Image Acquisition and Preprocessing

Remote sensing images used in this study were obtained from the Google Earth Engine (GEE) platform. Remote sensing images in 2006 and 2010 were from Landsat 5 ETM. Remote sensing images in 2016 and 2020 were from Landsat 8 OLI/TIRS. The spatial resolution of all the images was 30 m. Remote sensing images with the minimum cloud amount were selected in the GEE platform and then radiometric correction, image fusion, and other image preprocessing operations of Landsat images were completed.

### 2.4. Salt-Leaching Irrigation Information Extraction

In the Hetao Plain, water is irrigated to leaching saline soil in the autumn from the end of October to the beginning of November each year. The temporary water bodies are formed in the surface that did not exist at other times of the year. After extracting water bodies during the period and subtracting those long-term stable water bodies, such as rivers and lakes, the distribution of salt-leaching irrigation water information can be obtained. Salt-leaching irrigation can transport partial soil salts and some ions into groundwater with irrigation water or transport to adjacent areas [3,11,17]. To analyze the relationships among the distribution of salt-leaching irrigation, saline-alkali land, and groundwater arsenic concentration, we extracted the information of salt-leaching irrigation water in autumn of the previous year (Period I corresponding to 2005–2009 and Period II corresponding to

2015–2019) (Figure 3a). We observed that the salt-leaching irrigation position moderately change in each autumn, respectively, within the two periods. Therefore, we extracted the information about salt-leaching irrigation water in 2005, 2009, 2015, and 2019. To extract the distribution of the salt-leaching irrigation water with high accuracy from the remote sensing images, the Automated Water Extraction Index (*AWEI*) [40] was utilized for extracting water information based on Equations (1) and (2):

$$AWEI_{nsh} = 4 \times \left( \rho_{(green)} - \rho_{(SWIR1)} \right) - \left( 0.25 \times \rho_{(NIR)} + 2.75 \times \rho_{(SWIR2)} \right) \qquad (1)$$

$$AWEI_{sh} = \rho_{(blue)} + 2.5 \times \rho_{(green)} - 1.5 \times \left( \rho_{(NIR)} + \rho_{(SWIR1)} \right) - 0.25 \times \rho_{(SWIR2)} \qquad (2)$$

where $AWEI_{nsh}$ can effectively remove pixels without a water body, including dark-built surfaces in areas with urban background, and $AWEI_{sh}$ primarily further eliminates shadow pixels that $AWEI_{nsh}$ may not effectively remove. $\rho_{(blue)}$ refers to the gray value in the blue band of the images, $\rho_{(green)}$ is the gray value in the green band of the images, $\rho_{(NIR)}$ is the gray value in the near-infrared band of the images, $\rho_{(SWIR1)}$ refers to the gray value in the shortwave infrared 1 band of the images, and $\rho_{(SWIR2)}$ is the gray value in the shortwave infrared 2 band of the images. The surface water was extracted by the application of a threshold close to that in [40].

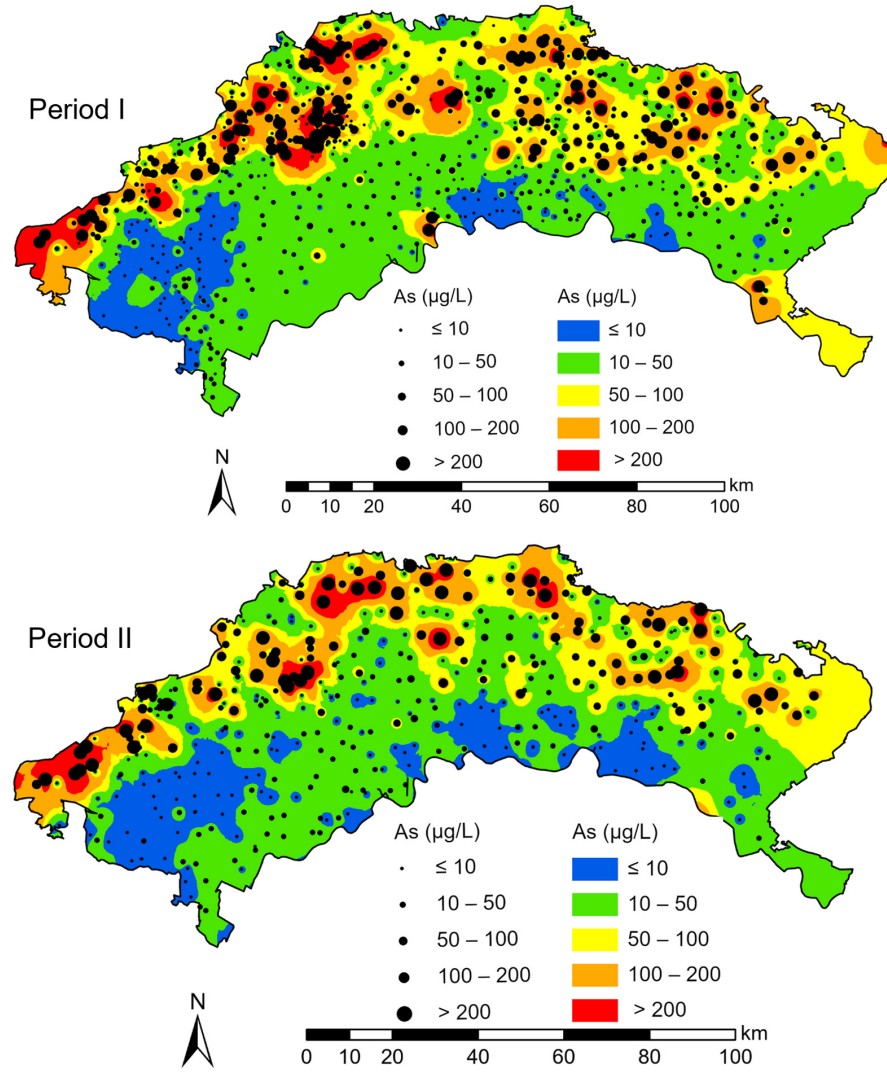

**Figure 2.** Spatial distribution of arsenic concentrations in sampling sites and in the IDW interpolation method.

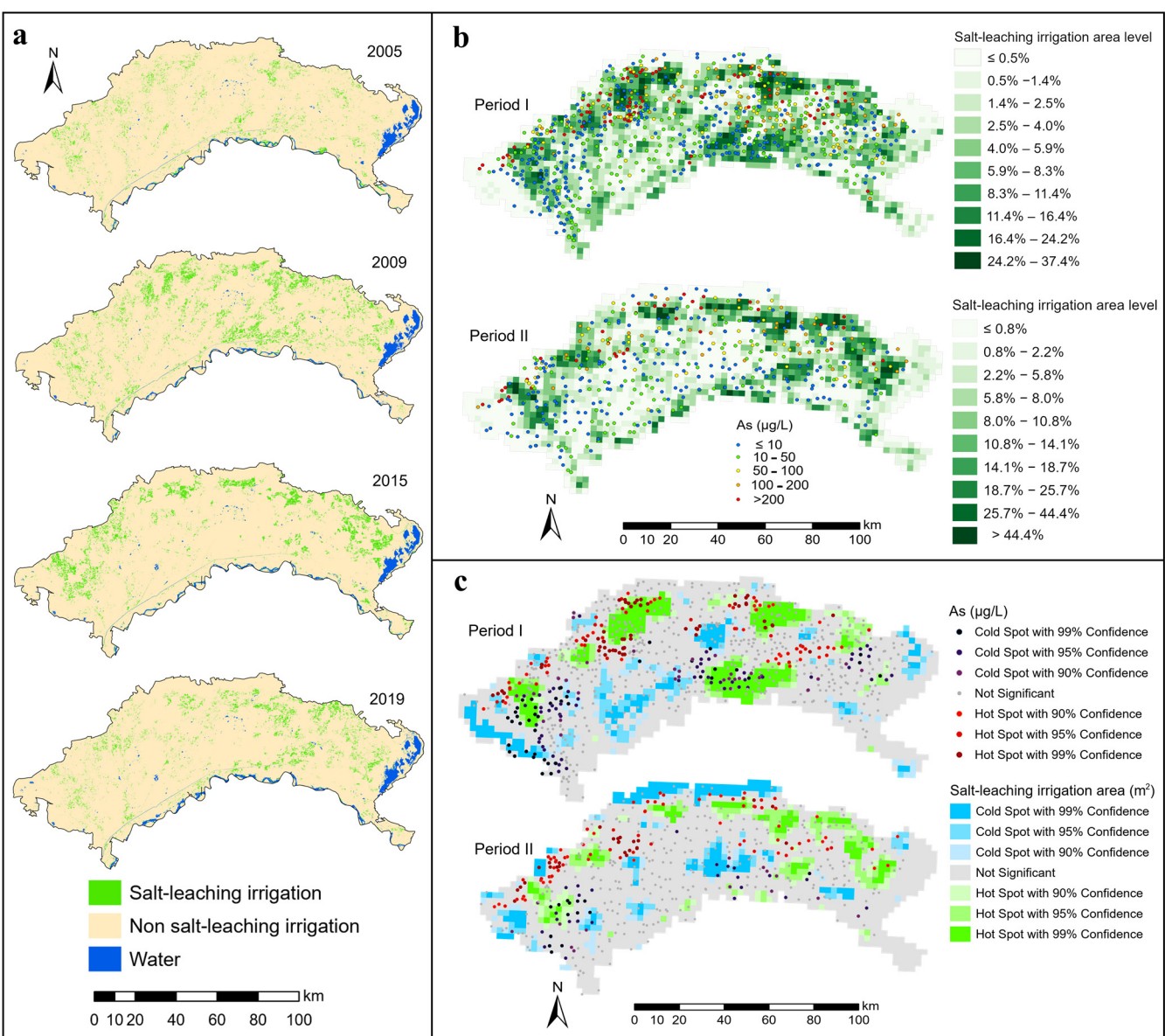

**Figure 3.** Spatial distribution and hotspots of salt-leaching irrigation and groundwater arsenic concentrations ((**a**) Salt-leaching irrigation information, (**b**) Covering area percentages of salt-leaching irrigation and groundwater arsenic concentrations, (**c**) Hotspots of salt-leaching irrigation area and groundwater arsenic concentrations).

## 2.5. Saline-Alkali Land Information Extraction

Supervised classification (random forest algorithm) was applied to obtain saline-alkali land information. The training samples used in the supervised classification were obtained according to a field survey and high-resolution images on Google Earth. The variation in land cover caused by seasonal changes affects the precision of the saline-alkali land interpretation. Therefore, the spring image was selected for saline-alkali land research when the amounts of vegetation and soil water content at the surface were lower [33]. Saline-alkali land varied slightly between 2006–2008, 2008–2010, 2016–2017, and 2018–2020. Therefore, we extracted information on saline-alkali land in 2006, 2010, 2016, and 2020 (Figure 4a). The land use of the study area was divided into three types: saline-alkali land (200–250 sample points), non-saline-alkali land (200–250 sample points), and water (100–150 sample points). In the Google Earth Engine (GEE) platform, the sample points were randomly separated into 7:3, which were used for training and verification sample

points of the land use classification algorithm. The classified products and sample points were then analyzed using the confusion matrix analysis. The overall accuracy and Kappa accuracy were calculated to assess the classification effect or misclassification error of the various classification algorithms. The overall accuracy remained in the range of 92–97%, and the Kappa accuracy remained in the range of 91–96%.

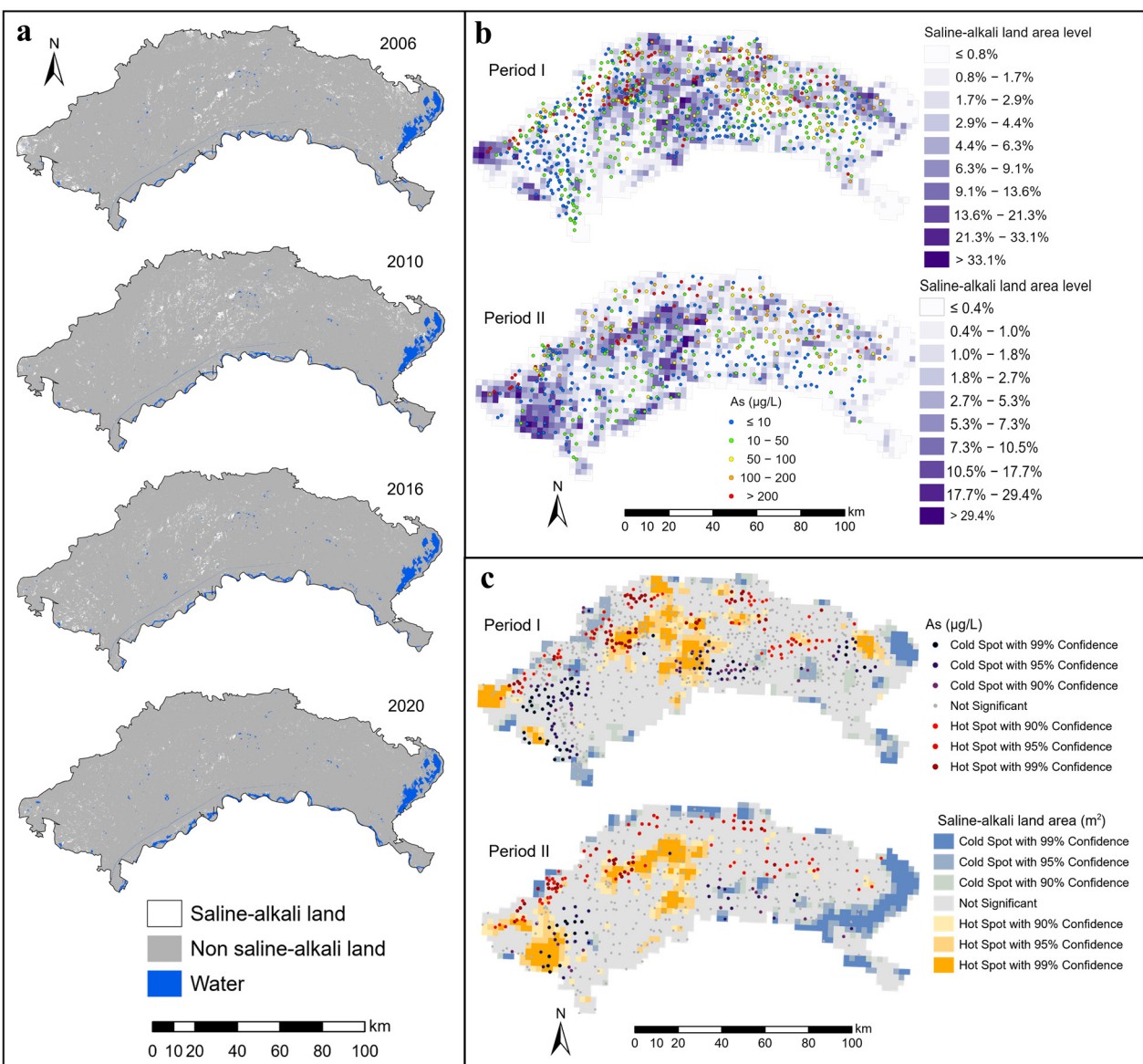

**Figure 4.** Spatial distribution and hotspots of saline-alkali land and groundwater arsenic concentrations ((**a**) Saline-alkali land information, (**b**) Covering area percentages of saline-alkali land and groundwater arsenic concentrations, (**c**) Hotspots of saline-alkali land area and groundwater arsenic concentrations).

## 2.6. Data Analysis Methods

**Statistics and classification:** The study area was divided into 1505 evaluation units of 2.5 × 2.5 km fishnets, and the covering area percentages of salt-leaching irrigation and saline-alkali land in each fishnet were calculated. Subsequently, the covering area percentages of salt-leaching irrigation and saline-alkali land in fishnets were classified into ten levels using the Natural Breaks Jenks method. The classification results were recorded as 'salt-leaching irrigation area level' and 'saline-alkali land area level' in Figures 3b and 4b. The Natural

Breaks Jenks Method can minimize the variance within classes and can maximize the variance between classes to make the classification most appropriate [41].

**Hotspot analysis**: Getis-Ord Gi* was used to calculate the hotspots and cold spots of the salt-leaching irrigation area, saline-alkali land area, and groundwater arsenic concentration in this study (Figures 3c and 4c). A high z-score refers to the hotspot, while a low z-score represents the cold spot. Before hotspot analysis, the K-S (Kolmogorov–Smirnov) test was applied to judge whether the data conform to the normal distribution. For the data that do not conform to normal distribution, Box-Cox transformation was performed. Hotspot analysis was conducted in the spatial statistics toolbox of ArcGIS Pro 2.8. The Gi* was calculated as [42]:

$$G_i^* = \frac{\sum_{j=1}^n \omega_{i,j} x_j - \overline{X} \sum_{j=1}^n \omega_{i,j}}{S\sqrt{\frac{n\sum_{j=1}^n \omega_{i,j}^2 - \left(\sum_{j=1}^n \omega_{i,j}\right)^2}{n-1}}}$$

where $x_j$ is the attribute value of $j$, $\omega_{i,j}$ refers to the spatial weight among feature $i$, $n$ is the total number of features, $\overline{x}$ is the mean value of all element values, and $S$ is the standard deviation of all element values.

**Path analysis**: A path model was used to evaluate the relationships among salt-leaching irrigation area, saline-alkali land area, As, and other ions' concentrations in groundwater. The model shows the direct and indirect effects of single factors and interactive factors on the receptor through multiple regression [43]. Before establishing a path model, the K-S (Kolmogorov–Smirnov) test was applied to determine whether the data conformed to the normal distribution. Box-Cox transformation was performed if the data did not conform to normal distribution. Path analysis was established in SPSSAU and was added or deleted by the significance of test parameters of each path ($p$-value < 0.05) to debug the model.

## 3. Results

### 3.1. Spatiotemporal Variations in Groundwater as Concentrations

In Period I, the As concentrations in the groundwater samples ranged from 0 to 999 μg/L. The mean and median concentrations were 76.90 and 21.60 μg/L, respectively. Approximately 64% of the total groundwater samples had As concentrations higher than 10 μg/L, and approximately 30.7% of the total samples had As concentrations higher than 50 μg/L. Figure 2 shows the extrapolation map of groundwater As using the inverse distance weighting (IDW) interpolation method. The groundwater samples with As concentrations higher than 50 μg/L mostly occurred along the northern edge of the study area adjacent to northern mountainous areas.

In Period II, the mean and median As concentrations were 67.85 and 14.80 μg/L, respectively, and the As concentrations ranged from 0.05 to 946.20 μg/L. Groundwater samples with As concentrations higher than 10 and 50 μg/L accounted for approximately 58% and 29.7% of the total groundwater samples, respectively. Figure 2 shows that the spatial distribution of groundwater As varied slightly between Periods I and II. Higher As concentrations in the groundwater were mainly found along the northern edge of the study area. The mean groundwater As concentration in Period II from the study area was slightly lower than that in Period I, whereas there was no obvious change in the spatial distribution trend of groundwater As between Periods I and II. In addition, areas with As concentrations higher than 10 μg/L decreased. The area with an As concentration in the range of 10–50 μg/L increased in the mid-west and southeast regions. The figure shows that the area with As concentrations higher than 50 μg/L expanded between Periods I and II.

### 3.2. Relationship between Salt-Leaching Irrigation Area Distribution and Groundwater as Concentrations

The results of the Automated Water Extraction Index procedure suggested that salt-leaching irrigation was mainly distributed along the front of the northern mountainous region and the areas along the Yellow River (Figure 3a). Salt-leaching irrigation distribution

was widespread in other areas but sparse in the central areas, as indicated by hot and cold spots in the salt-leaching irrigation area. Salt-leaching irrigation hotspots occurred in the northern edge of the study area and some areas in the south (Figure 3c). Many farmlands are located in these areas. Autumn irrigation is an efficient way to control soil salinity in farmlands by leaching salts from the root zone to non-irrigated adjacent areas under significant lateral groundwater gradients [3].

From the perspective of spatial location relationships, cold spots of groundwater As were more likely to be found in the hotspots of the salt-leaching irrigation area located near the Yellow River, while hotspots of groundwater As were more likely to be found in the hotspots of the salt-leaching irrigation area near the northern mountainous region (Figure 3c). The quantitative relationship between the salt-leaching irrigation area level (SLIAL) and the proportion of As concentrations exceeding 10 μg/L was estimated by spatial overlay analysis. The results are shown in Figure 5. With the increase in SLIAL, the proportion of As concentrations exceeding 10 μg/L increased marginally and then fluctuated. Figure 6a shows the annual variation in As concentration in groundwater corresponding to the salt-leaching irrigation distribution. Disappeared salt-leaching irrigation refers to the regions where salt-leaching irrigation land was observed in Period I and did not appear in Period II. Newly increased salt-leaching irrigation refers to the regions where salt-leaching irrigation land did not appear in Period I and has newly appeared in Period II. The results of the Mann–Whitney test indicated that there was no significant difference in total As concentrations in groundwater between the disappeared salt leaching irrigation and the newly increased salt-leaching irrigation from Period I to Period II.

### 3.3. Relationship between Saline-Alkali Land Distribution and Groundwater as Concentrations

The results of the supervised classification revealed that the saline-alkali land was concentrated in the western, central, and northeastern regions during Period I (Figure 3). Saline-alkali land had the highest spatial aggregation with a contiguous chunk distribution. From Period I to Period II, the saline-alkali land area increased in the west and the degree of spatial aggregation also increased, while saline-alkali land changed from widespread distribution to fragmented distribution in the central area. The distribution of saline-alkali land gradually fragmented from Period I to Period II across the Hetao Plain. In the region with more salt-leaching irrigation areas, the areas of saline-alkali land were smaller (Figures 3 and 4). The hotspots of the saline-alkali land area are more likely to appear in the areas near the northern mountainous region (Figures 3c and 4c).

From the perspective of spatial location relationships, hotspots of groundwater As were more likely to occur in the areas near the northern mountainous region (Figure 4c), where groundwater flows converge [38]. The quantitative relationship between the saline-alkali land area level (SALAL) and the proportion of As concentrations exceeding 10 μg/L was estimated by spatial overlay analysis. The results are shown in Figure 5. The proportion of As concentrations exceeding 10 μg/L generally increased and then decreased with the increase in SALAL both in Period I and Period II (Figure 5). During Period I, the proportion of As concentrations exceeding 10 μg/L increased when SALAL was less than 7, and the proportion of arsenic concentrations exceeding 10 μg/L decreased when SALAL exceeded 7. However, in Period II, the proportion of arsenic concentrations exceeding 10 μg/L increased when SALAL was lower than 2, and the proportion of arsenic concentrations exceeding 10 μg/L decreased when SALAL exceeded 2. Figure 6b shows the annual variation in groundwater arsenic concentrations in correlation with the saline-alkali land distribution. The results showed that there was no significant difference in total arsenic concentration between the disappeared saline-alkali land and the newly increased saline-alkali land from Period I to Period II.

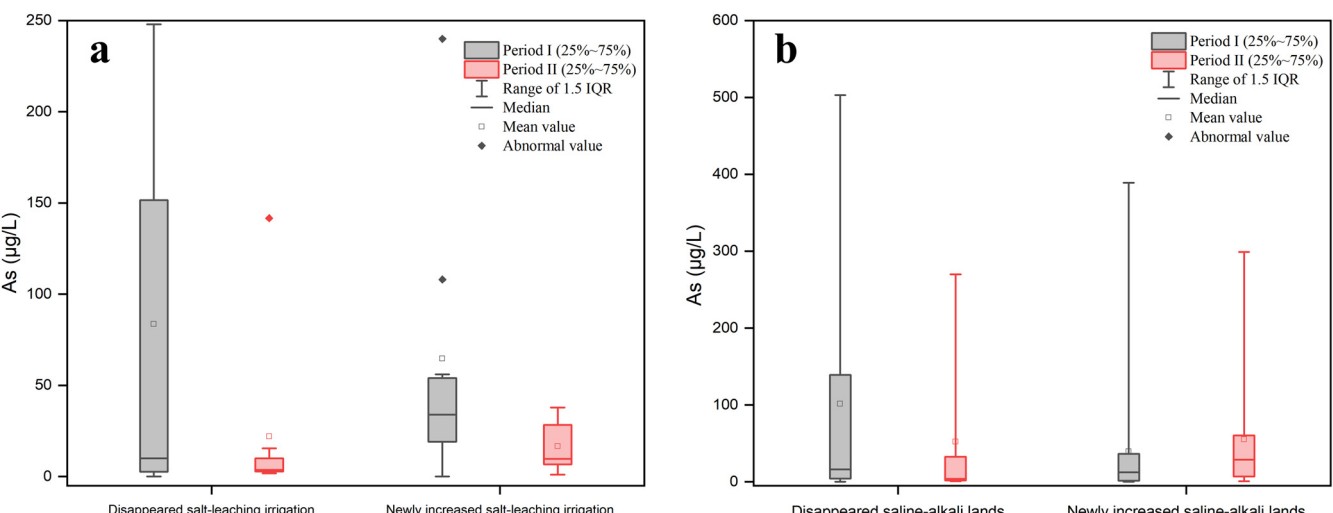

**Figure 5.** Relationships between the proportion of As concentrations in the groundwater exceeding 10 μg/L with the salt-leaching irrigation area level (SLIAL) and the saline-alkali land area level (SALAL).

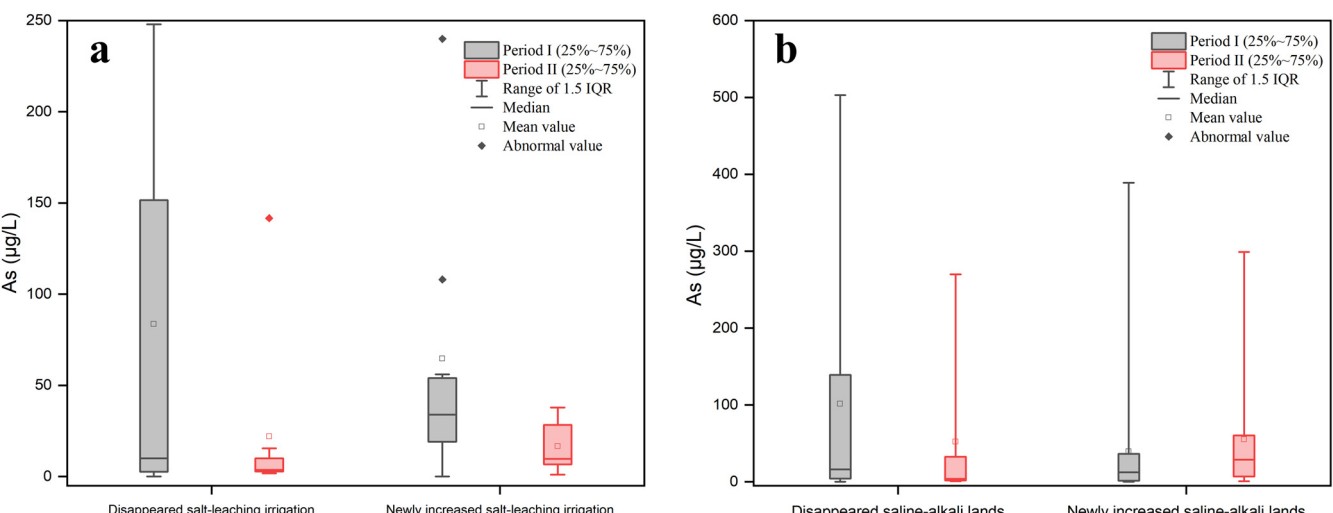

**Figure 6.** Annual variation in As concentrations in groundwater with salt-leaching irrigation and saline-alkali land distribution ((**a**) Annual variation in As concentrations in groundwater with salt-leaching irrigation, (**b**) Annual variation in As concentrations in groundwater with saline-alkali land distribution).

## 4. Discussion

### 4.1. The Impact of Salt-Leaching Irrigation and Saline-Alkali Land on as Transport

Flooding usually induces anoxic conditions that promote the release of As from soil and sediment [44]. However, seasonal short-term flooding may not considerably enhance As mobilization [45]. The proportion of As concentrations exceeding 10 μg/L may be enhanced with increasing SLIAL. In addition, salt-leaching irrigation usually increases the contents of $Na^+$, $Mg^{2+}$, $Ca^{2+}$, $HCO_3^-$, and $SO_4^{2-}$ in groundwater and decreases the contents of $Ca^{2+}$, $SO_4^{2-}$, $Na^+$, and $Cl^-$ in the soil of saline-alkali land [12]. Therefore, the distribution of saline-alkali land should be considered when exploring the impact of salt-leaching irrigation on As in groundwater.

During Period I, lower As concentrations generally occurred in areas with higher SLIAL and lower SALAL, accompanied by a higher proportion of As (III) (Figure 7). The large ratio of $n(Na^+)/n(Cl^-)$ (>0.86) indicates that the salinity in groundwater originated from leaching rather than sedimentation [46,47]. Figure 8 shows a larger ratio of $n(Na^+)/n(Cl^-)$, higher TDS concentration, and higher $SO_4^{2-}$ concentration in areas with higher SLIAL and lower SALAL, implying that salinity in groundwater in these areas is mainly derived from leaching saline soils. Areas with higher SLIAL and lower SALAL values suggested that salt-leaching irrigation in the previous year transferred salt from the soil into the groundwater. This decreased the redox condition of the groundwater. Less reductive conditions of high-saline groundwater (TDS > 3000 mg/L) prevented As enrichment with the reduction of $SO_4^{2-}$ [24]. A simultaneously higher proportion of As (III), higher TDS, and higher $SO_4^{2-}$ in the areas with higher SLIAL and lower SALAL implied that salt-leaching irrigation carried the soil salt ions into groundwater, limiting As enrichment in groundwater by decreasing reductive conditions and preventing the participation of $SO_4^{2-}$ in the redox process. Therefore, less saline-alkali land associated with more salt-leaching irrigation may decrease the arsenic concentration in groundwater.

Higher As concentrations tended to occur in areas with higher SLIAL and SALAL, accompanied by a relatively higher proportion of As (III) (Figure 7). Moreover, a lower ratio of $n(Na^+)/n(Cl^-)$ (<0.86), lower TDS concentration (TDS < 3000), and lower $SO_4^{2-}$ concentrations in the groundwater were observed in these areas. This suggests that more salt-leaching irrigation in the previous year did not transfer more salt from the soil to the groundwater in these areas. These areas also showed higher salt-leaching irrigation in the previous year, while saline-alkali land areas were not reduced. There could be serious soil secondary salinization phenomena in which groundwater salt is brought to the surface soil in the spring after salt-leaching irrigation in these areas. Irrigation can result in an increase in the groundwater table, and soluble salts from groundwater easily pass through the vadose zone and accumulate in shallow soil when the water table is relatively high [5]. A relatively high groundwater table can enhance groundwater reduction conditions [23]. A higher proportion of As (III) and lower $SO_4^{2-}$ concentrations also indicate higher reduction conditions for groundwater from these areas. As enrichment occurred simultaneously with the reduction of $SO_4^{2-}$ in the reductive groundwater with TDS below 3000 [24]. Therefore, a synergistic increase in salt-leaching irrigation and saline-alkali land may increase the As concentration in the groundwater.

Slightly higher groundwater As was found in areas with lower SLIAL and higher SALAL, while the proportion of As (III) decreased (<40%) (Figure 7). The pH and $HCO_3^-$ contents in groundwater from these areas were relatively higher (Figure 8). In relatively oxidized groundwater, higher pH and $HCO_3^-$ concentrations usually promote As desorption in groundwater [27,48]. In addition, areas with lower and higher SALAL were more likely to have low-lying downstream discharge areas [10]. Significant lateral movement of groundwater and transportation of salt were observed during the salt-leaching irrigation period, inducing the movement of water and salts from the root zone to adjacent areas [3]. With a decrease in elevation and groundwater flow, salt ions moved downstream to the discharge zone, causing the discharge zone to be more likely to be saline-alkaline land [10]. In the discharge areas, strong evaporation enhanced the weathering of carbonate

and increased the $HCO_3^-$ concentration and pH in groundwater, which promoted the desorption of As in groundwater [25,48], indicating that saline-alkali land may increase the As concentration in groundwater.

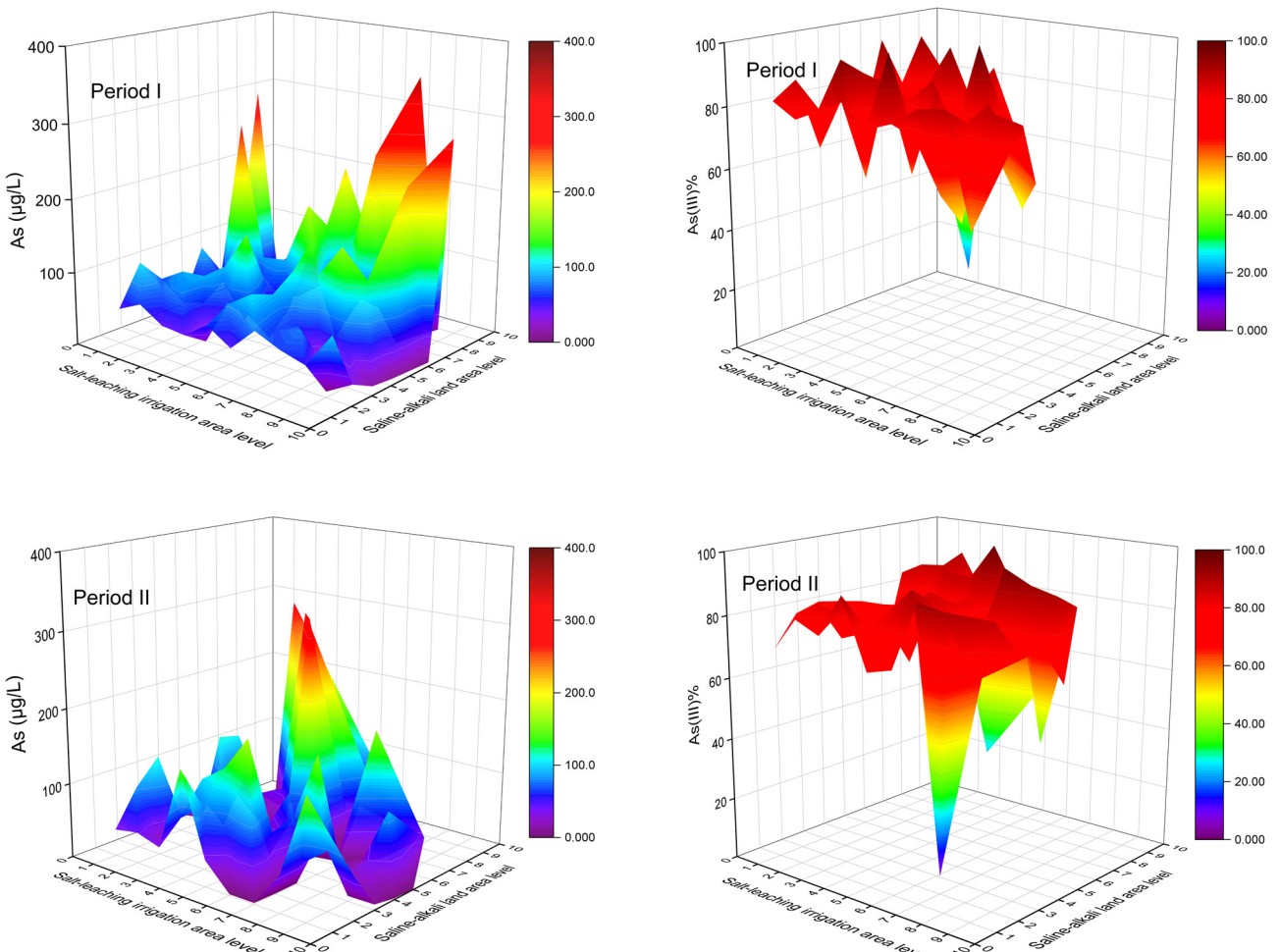

**Figure 7.** The variations of total arsenic concentrations and proportion of As (III) concentrations to total arsenic concentrations under the combined influence of SLIAL and SALAL.

In Period II, lower groundwater As levels were also observed in areas with higher SLIAL and lower SALAL. However, higher groundwater As levels were more likely to occur in areas with lower SLIAL and higher SALAL (Figure 7). Higher pH, TDS concentration, $SO_4^{2-}$ concentration, and a lower proportion of As (III) (<40%) in groundwater were also observed in these areas. Higher TDS concentrations (>3000 mg/L) occurred in areas with higher SLIAL and SALAL (Figure 8). This phenomenon might be associated with the decrease in the saline-alkali land area from Period I to Period II. From 2011 to 2016, the large-scale implementation of water-saving reconstruction in irrigation areas of the Hetao Plain considerably reduced the groundwater table [40]. With a decrease in the groundwater table, fewer groundwater salts returned to the surface soil, and the saline-alkali land area gradually decreased [13], possibly elevating salinity in groundwater. The increase in groundwater salinity usually weakens the reductive dissolution of As but promotes the desorption of As via competing adsorption with $HCO_3^-$ [11,24,28,49]. Consequently, the decrease in saline-alkali land area from Period I to Period II may have increased the desorption of As in groundwater.

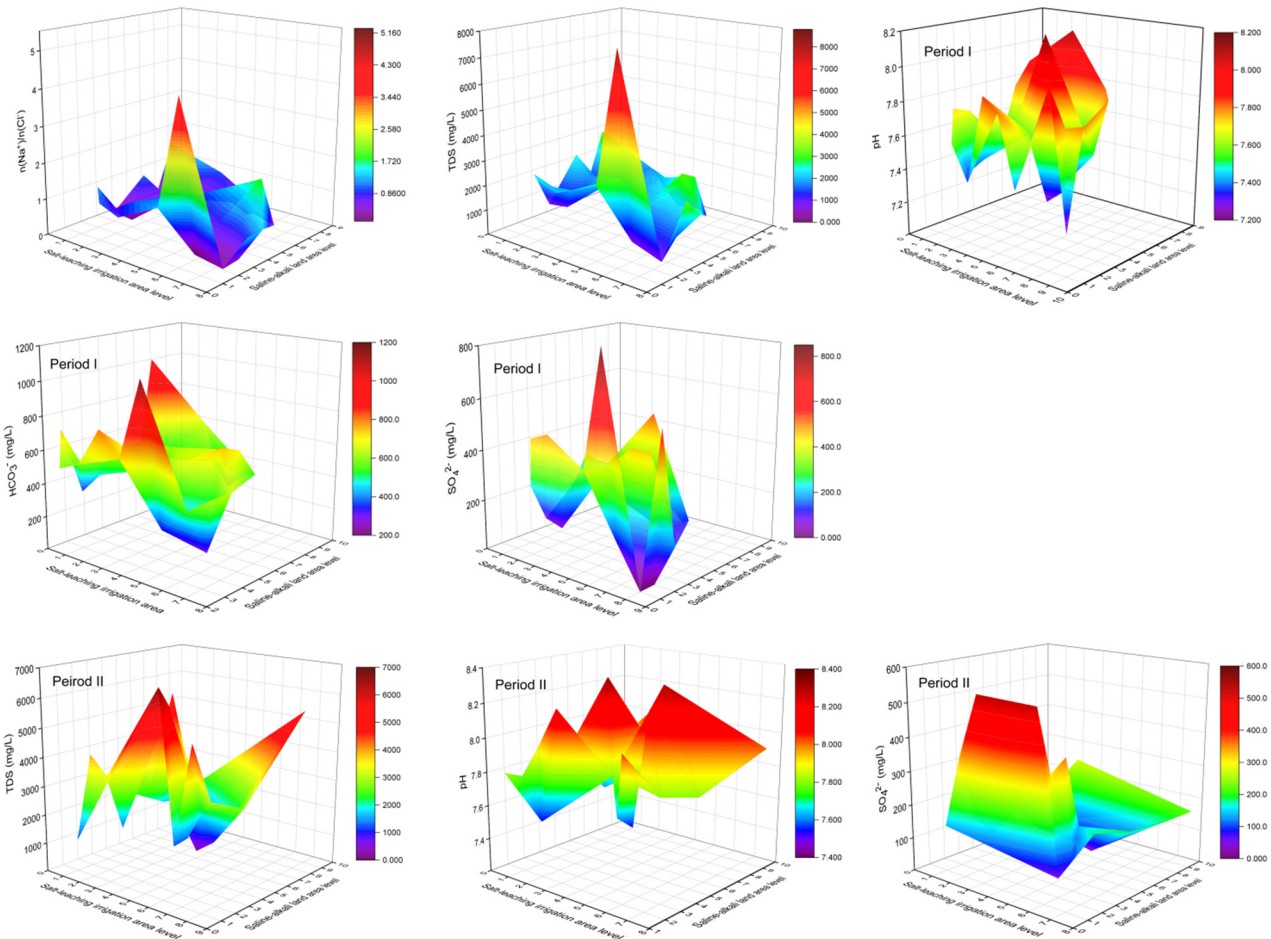

**Figure 8.** The variations of n(Na+)/n(Cl⁻) ratio, TDS, pH, $SO_4^{2-}$, and $HCO_3^-$ under the combined influence of SLIAL and SALAL.

The reductive dissolution of As-rich Fe (oxyhydr)oxide is the dominant reaction for As transport in groundwater, and As desorption generated by alkaline pH and increased $HCO_3^-$ is another significant process of As transport in the Hetao Basin [25]. Relative contributions of the two geochemical processes largely depend on redox conditions, pH, and competing anion concentrations in groundwater. Therefore, the covered area of salt-leaching irrigation and saline-alkali land may influence groundwater As mobilization through different geochemical processes. A path analysis model was used to estimate the influence path of salt-leaching irrigation and saline-alkali land on groundwater As mobilization across the Hetao Plain (Figure 9).

The results showed that the salt-leaching irrigation area was negatively correlated with the saline-alkali land area in both Periods I and II. In Period I, the salt leaching irrigation area negatively affected the $HCO_3^-$ concentration, while the $HCO_3^-$ concentration positively affected the As concentration in the groundwater. Previous research has revealed that increasing the $HCO_3^-$ concentration and pH promotes the release of As in groundwater due to competing adsorption [25]. Therefore, salt-leaching irrigation may restrain the release of As. The saline-alkali land area showed a negative correlation with the $SO_4^{2-}$ concentration, while the $SO_4^{2-}$ concentration showed a negative relationship with the As concentration in the groundwater. Decreased levels of $SO_4^{2-}$ promote As enrichment in groundwater [10,22]. This suggests that saline-alkali land can increase the As content in groundwater. In Period II, the saline-alkali land area showed a direct negative effect on the TDS concentration and an indirect negative effect on the $SO_4^{2-}$ concentration. TDS content was directly negatively associated with the As concentration. Generally,

a higher TDS content limits the enrichment of As in groundwater from Rachna Doab, Punjab Pakistan, and the Datong Basin, China [10,11]. However, the $SO_4^{2-}$ concentration showed no significant effects on the As concentration. This variation may be related to the decrease in saline-alkali land area from Period I to Period II due to water-saving irrigation in the study area, which increased the TDS concentration and enhanced the oxidization of groundwater. The results implied that salt-leaching irrigation can partially flush soil salinity into groundwater to reduce the saline-alkali land area and elevate the ion concentration in groundwater. Salt-leaching irrigation considerably elevated the contents of $Na^+$, $Mg^{2+}$, $Ca^{2+}$, $HCO_3^-$, and $SO_4^{2-}$ in groundwater [12]. These ions may affect As concentrations in the groundwater. We represent the above geochemical process of the impacts of salt-leaching irrigation and saline-alkali land on As transport in the adjacent shallow aquifers in Figure 10.

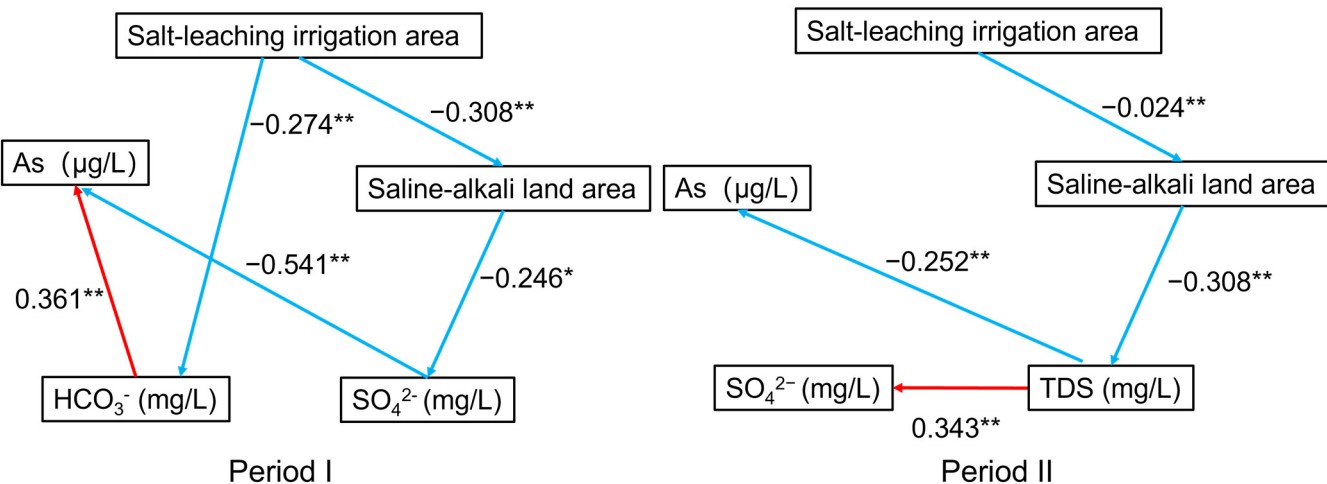

**Figure 9.** The results of path analysis (* $p < 0.05$, ** $p < 0.01$, red represents positive correlation and blue represents negative).

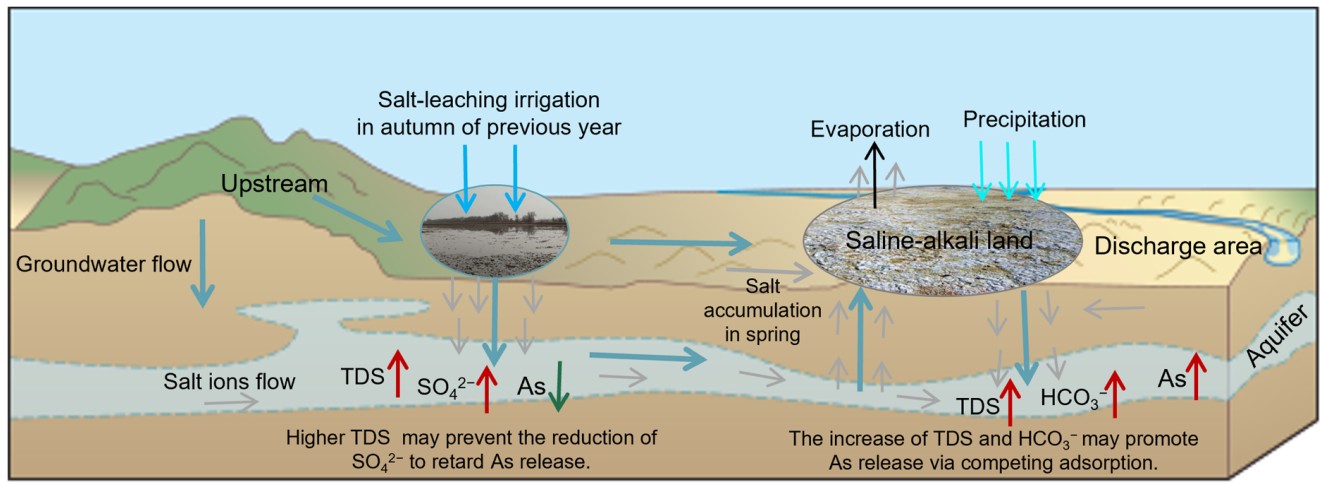

**Figure 10.** Schematic illustration of the impacts of salt-leaching irrigation and saline-alkali land on As transport in the adjacent shallow aquifers.

Previous studies had confirmed the effects of salt-leaching irrigation on the salinity and related salt ions' concentrations of saline-alkali land soil and groundwater, the impacts of saline-alkali land on salt ions' concentrations in groundwater, and the influence of groundwater salinity on As transport [3,8,12,14,34]. This paper has addressed this gap in the influence of the distribution of saline-alkali land and salt-leaching irrigation on

As transport in groundwater, which is not directly discussed in previous studies. Our results indicated that although salt-leaching irrigation is helpful to reduce the saline-alkali land area, the treatment of saline-alkali land may affect the control of arsenic pollution in groundwater. We suggest that the treatment of saline-alkali land should not only consider its influence on the water table and salt ions' concentrations, but also pay more attention to the mobilization of other pollutants caused by the changes in salt ions' concentrations in groundwater.

### 4.2. Limitations of the Study and Future Works

Arsenicosis was firstly recognized in 1990 in the center of Inner Mongolia [50]. Guo et al. found that Wuyuan and Alashan were highly contaminated areas [51]. Arsenic concentrations of 96.2% in water samples were above 50 µg/L in the Wuyuan area and 69.3% in the Alashan area [51]. The risk of arsenic pollution also existed in the arid and semi-arid reducing aquifers around the Hetao Plain. Arsenic concentration ranges changed from <1 to 1860 µg/L in the Huhhot Basin, <1.0 to 1300 µg/L in the Datong Basin, and <0.01 to 177 µg/L with an average concentration of 27.3 µg/L in the Yinchuan Basin [30,50]. A more careful comparison study at a larger spatial scale and in past times about the harmful parameters around the Hetao Plain should be further undertaken. This study quantified the impact of the distribution of saline-alkali land and salt-leaching irrigation on groundwater As migration. For the convenience of analysis, the salinization degree of saline-alkali land is not classified in this paper because salt-leaching irrigation is difficult to be classified according to the amount of irrigation water. It is ignored that the influence of saline-alkali land has different salinization degrees on groundwater As in this paper. Except for groundwater total arsenic concentrations, partial data of other ions were missing. For the influence of the evolution of saline-alkali land and salt-leaching irrigation on geochemical processes of groundwater As, preliminary guesses only have been made based on the trends and correlations between ion concentrations in groundwater. For future studies, the impact of the evolution of saline-alkali land and salt-leaching irrigation on groundwater arsenic geochemical processes needs to be further quantified through long-term observations and experiments. Moreover, our study also suggested that the effects of salt-leaching irrigation and saline-alkali land distribution on groundwater As concentrations are not linear due to the diverse effects of groundwater salinity on groundwater arsenic. Advanced models and methods are needed to simulate a reasonable range in the water amount and area of salt-leaching irrigation, which is expected to contribute to groundwater arsenic elimination in the future.

### 5. Conclusions

The results of this study indicate that salt-leaching irrigation and saline-alkali land influence the As concentration in groundwater. In general, the groundwater As concentration fluctuated slightly, with an increase in the SLIAL and with low SALAL. Likely, seasonal short-term flooding may not markedly enhance As mobilization. Lower groundwater As concentrations appeared in the regions with high SLIAL and low SALAL due to the increase in TDS generated by leaching soil salt into the groundwater, which may retard the accumulation of As in groundwater. A larger saline-alkali land area (high SALAL) increased the groundwater As concentration due to the introduction of salt ions. This may be related to the increase in the $HCO_3{}^-$ concentration promoting As release via competing adsorption or the increase in the $SO_4{}^{2-}$ concentration enhancing As release in reductive groundwater with lower TDS. The path analysis model confirmed that salt-leaching irrigation may increase groundwater salinity by leaching soil salt into the groundwater to affect groundwater As. From Periods I and II, the decrease in the saline-alkali land area may impact As mobilization due to competitive adsorption caused by the increase in the TDS concentration in groundwater. Salt-leaching irrigation in the autumn is one of the effective ways to control soil salinity in the Hetao Plain, but our study showed that this measure had adverse effects on groundwater quality. We suggest that the current measures,

including policy and engineering to control soil salinity, should be changed to protect groundwater resources.

**Author Contributions:** Conceptualization, B.W.; methodology, software, data collection and analysis, S.Y., Y.T. and B.W.; chemical experiment, S.Y. and Q.W.; writing—review and editing, all authors; funding acquisition, B.W. and L.Y. All authors have read and agreed to the published version of the manuscript.

**Funding:** Funding for this study was provided by the National Natural Science Foundation of China (Grant No. 41977400), the Second Tibetan Plateau Scientific Expedition and Research Program (STEP) (Grant No. 2019QZKK0607), and the Central Government Guides Local Science and Technology Development Program (Grant Nos. 2021ZY0047, XZ202201YD0014C).

**Conflicts of Interest:** The authors declare that they have no known competing financial interest or personal relationships that could have appeared to influence the work reported in this paper.

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
