# Peer review of "Dynamics of Spatiotemporal Variation of Groundwater Arsenic Due to Salt-Leaching Irrigation and Saline-Alkali Land"

_remotesensing, doi:10.3390/rs14215586_

Round 1
Reviewer 1 Report
It is an interesting study about the relationship between As and other ions concertation and leaching irrigation. The field data is valuable and the analysis is instructive. There are some comments and suggestions.
(1) the quality of figures should be improved. It is not clear both on resolution and contents.
(2) Additional data are necessary to support the result and conclusion; they are the AS contents in sampling soil and leaching irrigation water.
(3) the path analysis of As with SALAL and SLIAL is not persuasive, as if there are some sentences and words about redox and ion’s reaction.
(4) what is the purpose of the two periods and how long the irrigation district has been irrigated? Comparing the As changes in the whole history with leaching irrigation and salinity land area, are the law findings in the two periods still there?
Reviewer 2 Report
The manuscript presented for review related to important issue aboutthe quality of the groundwater.
This research concerns the zone of low water abundance,
where there are significant differences in precipitation feed over the
hydrological year.
The studies presented are well explained. The discussion and conclusion part of the manuscript are clearly and understandable written.
The week part of the manuscript is a graphical section, which needs improvement. Below is a list of graphics that must be add/improve in case to publish the article in the MDPI remote sensing journal:
1) Due to the worldwide nature of the MDPI journals, please refer to the first graphic of a location sketch of a given location of the research area on a topographic map against the background of the country and continent. Otherwise, it is difficult for a foreign reader to get an idea of ​​the geomorphological position of the research area. This position is important for setting the precipitation supply of aquifers.
This article concerns with hydrogeology research.
The basic visualization method in hydrogeological research area are
geological maps and geological cross-sections including geological background,
permeability of layers, and groundwater level(piezometric head equipotential
line). I am asking you to add maps and cross-sections to manuscript such right now, and if there is no possibility please explain why such analyzes were not carried out. The presented results, without reference to the geological structure, and in particular to the permeability of the aeration zone, are not very reliable and may be subject to significant errors.
3) Graphics 3 and 4 are of very low quality, maps have a very low resolution, please replace the graphics in the final version of the article, intended for publication.
After introducing the above-mentioned additions, I recommends the article for publication.
Reviewer 3 Report
Comments are attached for the improvement of the manuscript after which the manuscript can be considered for publication.

Round 2
Reviewer 1 Report
The revision has responded all the concerning properly. Although there are still some points can be improved, the author mentioned it will be study in future. So, it is suggested to accept the manuscript as it is.
Author Response
Dear reviewer:
We would like to express our sincere appreciation to you for the work and time that you have spent reviewing our manuscript (entitled: ‘Dynamics of spatio-temporal variation of groundwater arsenic due to salt-leaching irrigation and saline-alkali land’). We have checked the English spelling and corrected the mistakes.
We hope that the new revision can be considered suitable for publication. If not, we will be glad to make any further changes that you may require.
Thank you and best regards.
Yours sincerely,
Wei Binggan
Reviewer 3 Report
Accept in present form
Author Response
Dear reviewer:
We would like to express our sincere appreciation to you for the work and time that you have spent reviewing our manuscript (entitled: ‘Dynamics of spatio-temporal variation of groundwater arsenic due to salt-leaching irrigation and saline-alkali land’). We have asked native English speakers to help us modify English language.
We hope that the new revision can be considered suitable for publication. If not, we will be glad to make any further changes that you may require.
Thank you and best regards.
Yours sincerely,
Wei Binggan